**RESEARCH**

# Beyond benchmarking and towards predictive models of dataset-specific single-cell RNA-seq pipeline performance

Cindy Fang[1,2,6], Alina Selega[1,3†] and Kieran R. Campbell[1,3,4,5*†]

[†]Alina Selega and Kieran R. Campbell jointly supervised this work.

*Correspondence:
kierancampbell@lunenfeld.ca

[1] Lunenfeld-Tanenbaum Research Institute, Toronto, Canada
[2] Program in Bioinformatics and Computational Biology, University of Toronto, Toronto, Canada
[3] Vector Institute, Toronto, Canada
[4] Departments of Molecular Genetics, Statistical Sciences, Computer Science, University of Toronto, Toronto, Canada
[5] Ontario Institute for Cancer Research, Toronto, Canada
[6] Present address: Department of Biostatistics, Johns Hopkins University, Baltimore, USA

## Abstract

**Background:** The advent of single-cell RNA-sequencing (scRNA-seq) has driven significant computational methods development for all steps in the scRNA-seq data analysis pipeline, including filtering, normalization, and clustering. The large number of methods and their resulting parameter combinations has created a combinatorial set of possible pipelines to analyze scRNA-seq data, which leads to the obvious question: which is best? Several benchmarking studies compare methods but frequently find variable performance depending on dataset and pipeline characteristics. Alternatively, the large number of scRNA-seq datasets along with advances in supervised machine learning raise a tantalizing possibility: could the optimal pipeline be predicted for a given dataset?

**Results:** Here, we begin to answer this question by applying 288 scRNA-seq analysis pipelines to 86 datasets and quantifying pipeline success via a range of measures evaluating cluster purity and biological plausibility. We build supervised machine learning models to predict pipeline success given a range of dataset and pipeline characteristics. We find that prediction performance is significantly better than random and that in many cases pipelines predicted to perform well provide clustering outputs similar to expert-annotated cell type labels. We identify characteristics of datasets that correlate with strong prediction performance that could guide when such prediction models may be useful.

**Conclusions:** Supervised machine learning models have utility for recommending analysis pipelines and therefore the potential to alleviate the burden of choosing from the near-infinite number of possibilities. Different aspects of datasets influence the predictive performance of such models which will further guide users.

**Keywords:** Single-cell RNA sequencing (scRNA-seq), Clustering, Benchmarking, Automated machine learning

## Background

Single-cell RNA-sequencing (scRNA-seq) has revolutionized biomedicine by enabling transcriptome-wide quantification of gene expression at single-cell resolution [1, 2]. The analysis of scRNA-seq data is often complex, requiring multiple interacting components such as cell filtering, normalization, dimensionality reduction, and clustering, the choices of which may affect the results of methods downstream. To meet this demand, there has been a surge in computational methods development, with over 1000 tools developed as of late 2021 and over 270 developed for cell clustering alone [3].

Concomitant with the development of this large toolset is the combinatorial number of possible pipelines that can be applied to a given scRNA-seq dataset, where a given pipeline is defined by the composition of methods for each step, along with their respective parameter choices. Consider an unrealistically simple example: if there are 3 analysis steps (e.g., filtering, normalization, clustering), with 4 computational methods for each step, and each method has 2 possible parameter combinations, then there are $(4 \times 2)^3 = 512$ possible pipelines. Given the far larger set of possibilities for steps, methods, and parameters, in practice, the number of sensible pipelines that could be applied to scRNA-seq data is likely in the high thousands if not millions. This therefore leads to an important question: how do we select the pipeline that is "best" for our dataset?

To tackle this, the field has largely relied on benchmarking studies. For example, numerous papers have performed comprehensive evaluations of multiple stages of the scRNA-seq workflow, including clustering [4], pseudotemporal ordering [5, 6], dimensionality reduction [7], dataset integration [8], data imputation [9], and gene selection [10]. These stages may be combined in frameworks such as pipeComp [11] that integrate multiple pipeline steps together to benchmark over combinations. While such benchmarking studies are hugely valuable, they may be limited for two reasons. Firstly, the extremely large combinatorial number of pipelines means it is impossible to exhaustively benchmark all method combinations, and so interactions between the different stages may affect performance. Secondly, pipeline performance is generally dataset-specific, meaning the pipeline that performs best on average may be not optimal for a given dataset.

This combinatorial number of possible pipelines is not unique to single-cell analysis. In supervised machine learning (ML), attempting to optimize the possible set of data processing pipelines, algorithms, and hyperparameter choices has led to the rise of automated machine learning (AutoML) [12]. AutoML algorithms attempt to automatically select optimal hyperparameter combinations using advanced statistical techniques such as Bayesian optimization [13]. Importantly, previous AutoML work has shown that rather than considering an exhaustive set of possible pipelines, testing over a large but fixed subset and recommending one from within that is sufficient [14]. However, in the context of supervised ML, the goal is to optimize the accuracy of predictive models on a held-out dataset which does not readily apply to single-cell analysis. In contrast, the majority of single-cell pipelines are unsupervised, and there are few methods developed at the intersection of AutoML and unsupervised genomic analysis [15].

Here, we investigate whether AutoML approaches may be adapted and applied for the optimization of scRNA-seq analysis pipelines in order to recommend an analysis pipeline for a given dataset. To do so, we applied 288 scRNA-seq clustering pipelines to 86

datasets, resulting in 24,768 unique clustering outputs, and quantified the performance of each via a range of cluster purity and gene set enrichment metrics. With this, we created *Single Cell pIpeline PredIctiOn (SCIPIO-86)*, the first dataset of single-cell pipeline performance. We then developed a set of supervised machine learning models to predict the performance of a given pipeline on a given dataset using a combination of pipeline and dataset features (Fig. 1A). We investigated the accuracy of predictions tailored to both datasets and pipelines and compared these to predictions that only use pipeline characteristics as input features. We further investigated the relevance of such predictions by correlating how the overlap of the identified clusters with existing cell labels varies for pipelines that were predicted by our model to perform well or not. Finally, we investigate what features of an scRNA-seq dataset make it easier to predict which

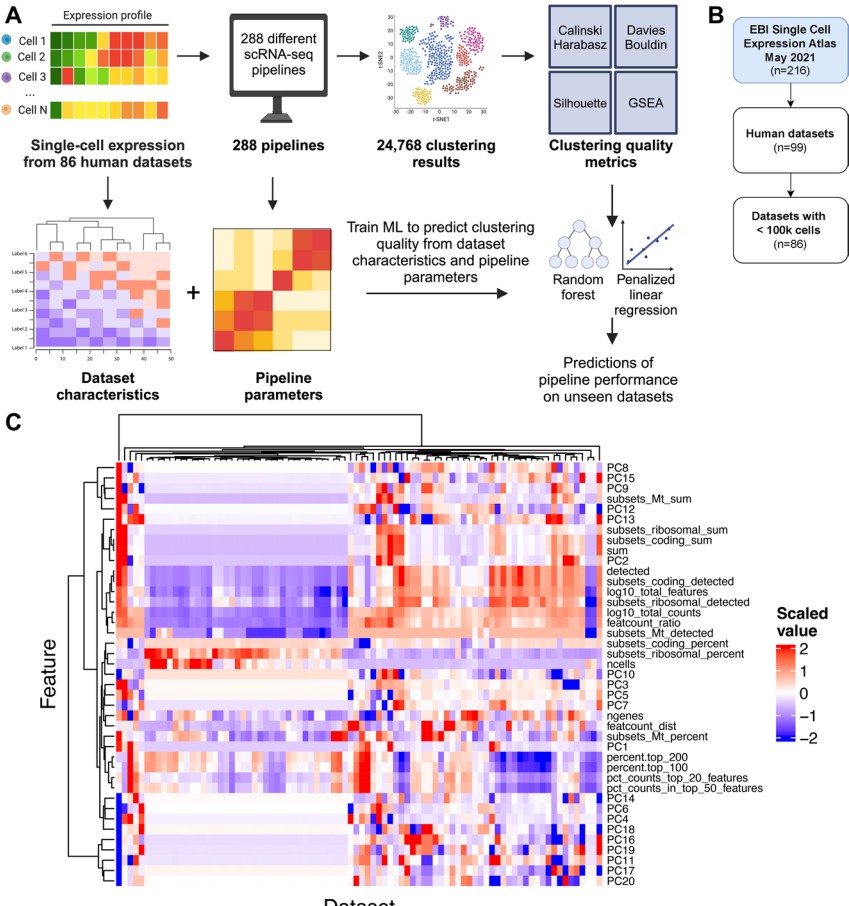

**Fig. 1** **A** Overview of the machine learning workflow: 288 clustering pipelines were run over each dataset and the success of each was quantified with 4 unsupervised metrics. Dataset- and pipeline-specific features were then computed and given as input to supervised machine learning models to predict metric values. **B** 86 human datasets in the EBI Single Cell Expression Atlas containing < 100 k cells as of May 2021 were selected for this study. **C** Characteristics of the 86 datasets used as input to predictive models of dataset-specific pipeline performance. These include median values of metrics frequently used to quality control at the cell level (e.g., percentage of mitochondrial counts) as well as principal components of average expression values per dataset jointly decomposed. Each characteristic was scaled in the training set to follow a standard normal distribution. The means and variances before scaling of each characteristic in the training set were used to scale the corresponding characteristics in the test set to prevent train-test leakage

pipelines will be applicable to it. This study provides a foundation for the development of recommender systems for scRNA-seq analysis pipelines, and the training data are publicly available at zenodo.org/records/11403435 to encourage further methods development in this area.

## Results

### Constructing a dataset of scRNA-seq pipeline performance on 86 datasets across 288 pipelines

We began by gathering a comprehensive pan-tissue pan-disease collection of 86 human scRNA-seq datasets (Additional file 2: Table S1) from EMBL-EBI's Single Cell Expression Atlas [16] comprising 1,271,052 cells total (Fig. 1B). For each dataset, we computed a wide range of dataset characteristics such as number of cells and proportion of genes detected (Additional file 2: Table S2 and the "Methods" section). Visualizing these based on the different dataset characteristics demonstrates significant heterogeneity between datasets with distinct clusters forming (Fig. 1C), in line with previous research [11].

Next, for each of the 86 datasets, we ran 288 different scRNA-seq analysis pipelines to produce a total of 24,768 unique clustering outputs using established frameworks [11]. The pipelines consisted of the four major steps in scRNA-seq clustering analysis: (i) filtering, (ii) normalization, (iii) dimensionality reduction, and (iv) clustering (see the "Methods" section). For each of the four steps, different algorithms and parameters were considered. For example, for normalization, we considered Seurat's log-normalization [17], scran's pooling-based normalization [18], and sctransform's variance-stabilizing normalization [19], which are three of the most popular normalization methods available as R packages [3]. In total, all possible parameter and algorithm combinations resulted in the 288 pipelines that were applied to each dataset.

Once the $86 \times 288$ clustering outputs were generated, we quantified the performance of each pipeline on each dataset using several different metrics. Since the majority of the scRNA-seq datasets did not include previous cell type annotations, we computed four unsupervised metrics for every pipeline-dataset pair (Fig. 2A) that target a measure of cluster purity or biological plausibility in line with previous benchmarking studies [8, 20]. The cluster purity metrics considered included the Calinski-Harabasz index (which from here we abbreviate to CH), Davies-Bouldin index (DB), and mean silhouette coefficient (SIL). CH measures the ratio of between-cluster dispersion to within-cluster dispersion, favoring well-separated, dense clusters. DB measures cluster similarity by comparing each cluster to its most similar one, attributing good scores to distinct, well-separated clusters. SIL measures how well each data point fits into its assigned cluster, with higher scores signifying consistent clusters (a point is closer to members of its own cluster) and negative scores suggesting misclassification. We opted for these three metrics as it was not a priori obvious if they would be correlated nor which one to favor above the others, and each may be computed for any clustering result in the absence of additional information such as cell labels. In addition, we used Gene Set Enrichment Analysis (GSEA) [21] to assess the biological plausibility of the clustering outputs, complementing the unsupervised metrics that rely on distances only. Ideally, each individual cluster should represent a biologically meaningful group of cells. Therefore, GSEA may be

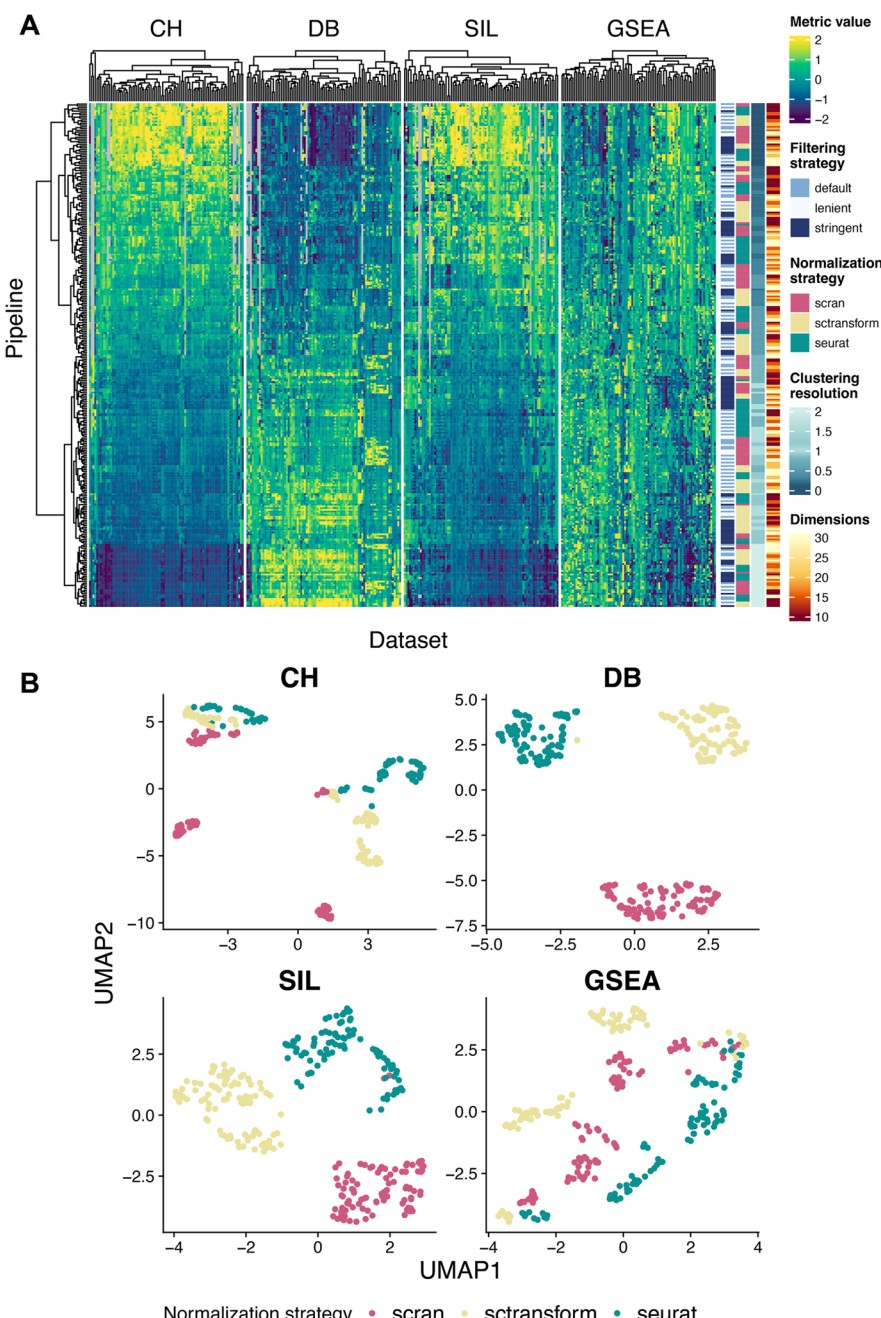

**Fig. 2** **A** The SCIPIO-86 dataset showing 288 scRNA-seq clustering pipelines applied to 86 datasets in terms of 4 unsupervised metrics (CH, DB, SIL, GSEA). Pipeline performance can be seen to have a monotonic relationship with clustering resolution for the CH, DB, and SIL metrics, with additional dependency on different pipeline strategies (filtering, normalization) also evident. **B** After correction for number of clusters, we embedded each scRNA-seq clustering pipeline into UMAP space, with variation visible due to normalization strategy

applied to differentially expressed genes between clusters to evaluate whether each cluster is enriched for Gene Ontology gene sets, which can be summarized over gene sets and clusters to a per-clustering score. While there are many possible ways

to construct such a score, we used the absolute normalized enrichment score (NES) averaged over clusters (see the "Methods" section).

We next observed that the three cluster purity metrics (CH, DB, SIL) exhibited a strong relationship with the number of clusters identified, typically showing a monotonically decreasing relationship with the number of clusters (Additional file 1: Fig. S1), similar to an observation made in a previous benchmarking study [11]. This represents a challenge for using such metrics to optimize scRNA-seq pipelines as most pipelines include a parameter for controlling the number of clusters (the resolution in graph-based clustering). Therefore, under this framework, such a parameter could be arbitrarily adjusted to achieve better performance. To counteract this effect, we trained a loess model for each dataset to regress out the number of clusters from each metric and established the residuals as the *corrected metric* (the "Methods" section). The corrected metrics (by construction) no longer showed the same relationship with the number of clusters and thus they allow for the quantification of pipeline accuracy in the absence of the confounding effect of number of clusters (Additional file 1: Fig. S2). Interestingly, since each pipeline is now represented by a high-dimensional vector of metrics over datasets, we can embed them into low-dimensional space using popular methods such as UMAP [22]. Now, each point represents an entire scRNA-seq pipeline rather than a cell and may be used to visually assess the results in a qualitative manner, which in our case shows the low-dimensional space being driven by normalization strategy (Fig. 2B).

Using corrected metrics to measure pipeline success, we found no single pipeline performed best across all datasets (Additional file 1: Fig. S3), and the best pipeline on average was not the same across each metric (Additional file 2: Table S3). Together, this represents the first dataset of single-cell pipeline performance comprising 4 corrected metrics across 24,768 dataset-pipeline pairs that we term the *Single Cell pIpeline PredIctiOn (SCIPIO-86)* dataset. Importantly, in line with previous findings [4], we found that the performance of the analysis pipelines were dependent on the dataset, providing additional motivation to model pipeline performance as a function of dataset-specific characteristics and pipeline parameters.

### Supervised models can predict dataset-specific pipeline performance

Next, we explored the ability of different supervised machine learning models to predict the four unsupervised metrics. We fitted random forest (RF) and penalized linear regression (LR) models with optimized hyperparameters tuned via cross-validation on the train set ($n = 61$, see the "Methods" section). For each model and metric, we considered two input feature sets: *dataset-pipeline interactions* and *pipeline features only*. In the *dataset-pipeline interactions* setting, interaction terms are allowed between dataset and pipeline features in addition to separate dataset and pipeline features in LR (and interactions naturally occur in RF), meaning pipeline performance predictions are dataset-specific. In the *pipeline features only* setting, the models have access to pipeline features only, resulting in predictions that are not dataset-specific.

The resulting predictive power of each model can be seen in Fig. 3 (top). Here, predictive power for a given test dataset is measured as a correlation between the predicted metric value for each pipeline on that dataset and the actual (calculated) value. This is

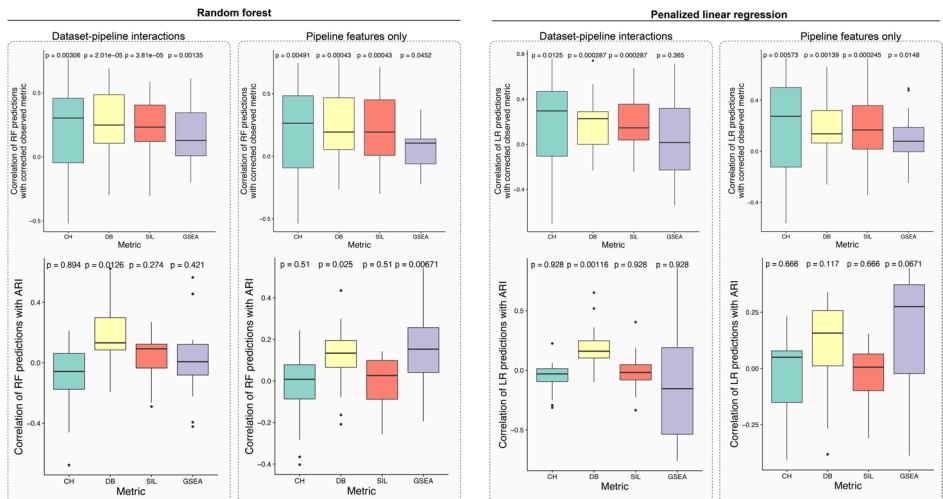

**Fig. 3** Test set (held out dataset) correlations of predicted unsupervised metric values with actual metric values (top) and pipeline-specific adjusted Rand index (ARI) (bottom) for random forest and penalized linear regression models, considering two settings of pipeline features only and dataset-pipeline interactions. In general, including dataset-pipeline interactions improves performance when predicting the unsupervised metrics (top) but degrades performance when predictions are contrasted to pipeline-specific ARI (bottom). *P*-values are Benjamini–Hochberg multiple test corrected

performed separately for each of the four metrics and for every dataset in the test set ($n=25$). For the RF model, prediction power was significantly ($p_\text{adj} < 0.05$, one-sided Wilcoxon rank-sum test with Benjamini–Hochberg multiple test correction) greater than random (0) in all settings. Interestingly, for all metrics, incorporating pipeline and dataset-specific features (resulting in dataset-specific recommendations) improved predictive performance in terms of median correlation (Additional file 2: Table S4). This pattern was largely similar when examining the results of the LR model: the average correlation was greater than 0 in all settings and significantly greater than 0 for 7/8 tests ($p_\text{adj} < 0.05$). For some metrics, LR prediction results improved when using dataset-pipeline interactions over pipeline features alone, either in terms of average correlation or *p*-value. Together, these results imply that (i) relatively simple supervised machine learning models can be used to predict the success of an scRNA-seq pipeline, (ii) incorporating dataset-specific features improves prediction accuracy, and (iii) these results are consistent for both RF and LR models, suggesting that they are a statement about the predictive capacity of the information contained in the data rather than exact machine learning model used.

Next, we investigated if the predictive performance of these models differs in atlas-scale datasets containing > 100,000 cells. We processed an additional 6 datasets (Additional file 2: Table S5) with the same pipelines as before (except for those using scran normalization that gave an out of memory error) and applied the trained predictive models from above to them. Correlating the predicted and actual metric values shows broadly comparable or improved performance (Additional file 1: Fig. S4), implying such predictive models will scale to larger and more complex datasets.

While the optimization of such metrics has previously been successfully used to benchmark single-cell pipelines [23], they are surrogate objectives that do not

necessarily favor pipelines that return more meaningful cellular populations. To address this, we exploited the fact that 16/25 test datasets had cell types/states previously annotated. While using existing annotations as "ground truth" has certain limitations that we cover in the discussion, it has previously been successfully used as an additional clustering comparator that provides a measure of how well a given clustering overlaps with one that is performed manually [24]. To quantify the overlap of a particular analysis pipeline with the previous labels, we computed the adjusted Rand index (ARI)—a measure of cluster overlap—between the clustering returned by every pipeline and the existing labels.

Next, we correlated this computed ARI with the predicted metric values across pipelines and datasets (the "Methods" section) with the results shown in Fig. 3 (bottom). For the RF model, when including dataset-pipeline interactions, 3/4 of the predicted metrics had positive average correlations with the ARI (1/4 significantly greater than 0 at $p_{adj} < 0.05$, one-sided Wilcoxon rank-sum test). An example may be seen in Additional file 1: Fig. S5 that displays UMAP plots of the clustering results of the best predicted pipeline in contrast with a randomly chosen pipeline, demonstrating that the pipeline recommended by our model more closely agrees with the expert annotations. In contrast, when not including dataset features, all four metrics had positive average correlations with ARI (2/4 significantly greater than 0 at $p_{adj} < 0.05$, one-sided Wilcoxon rank-sum test). A similar pattern is apparent in the results of the LR model, with the predictions using pipeline features achieving higher correlation with ground truth than those integrating dataset-pipeline interactions. Taken together, we interpret this as meaning that overall, the supervised ML models can use the prediction of surrogate metrics to recommend pipelines that correspond more closely to those optimized for the previous annotation of the single-cell data. However, when allowing for dataset-pipeline interactions (i.e., predictions "personalized" to each dataset), in most cases, there is no longer a significant association, implying that such models may be subtly overfit to the characteristics of the unsupervised metrics rather than truly meaningful cellular populations.

Next, we examined the qualitative differences between the best predicted pipeline and the Seurat version 5.0.3 default pipeline on two representative datasets from our test data for which author ground truth annotations were available. First, we examined an scRNA-seq dataset of whole blood from patients with COVID-19 and controls [25]. We found the best predicted pipeline learned 12 cell clusters that matched closer to the 6 cell types annotated by the authors compared with the 18 found by the Seurat default pipeline (Additional file 1: Fig. S6A). However, this may be simply due to the predicted pipeline learning fewer clusters overall. Therefore, we examined an additional dataset of scRNA-seq of colorectal tumors [26] where the Seurat default pipeline learns a number of clusters ($n = 26$) much closer to the annotated number ($n = 40$) than that returned by the best predicted pipeline ($n = 16$) (Additional file 1: Fig. S6B). Despite this, the default Seurat pipeline splits several annotated clusters into multiple subclusters, while the best predicted pipeline does not or does to a far lesser extent (Additional file 1: Fig. S6C).

Finally, we considered if additional metadata provided by each original study's authors would be of benefit for predicting dataset-specific pipeline performance. Given the large proportion of missing entries in each study's metadata (Additional

file 3), we focused on three that were near-universally available for all datasets: (i) the end-bias of the sequencing protocol (5′, 3′, or none), (ii) whether the cells had been FACS sorted (yes or no), and (iii) whether the cells originated from blood (yes or no). After retraining the machine learning models to predict dataset-specific pipeline performance incorporating these features, we found largely equivalent model performance (Additional file 1: Fig. S7).

### Dataset-specific characteristics impact predictive performance

Given that the predictive performance of each model on each metric varies across datasets (Fig. 3), we next investigated what dataset characteristics correlate with good or

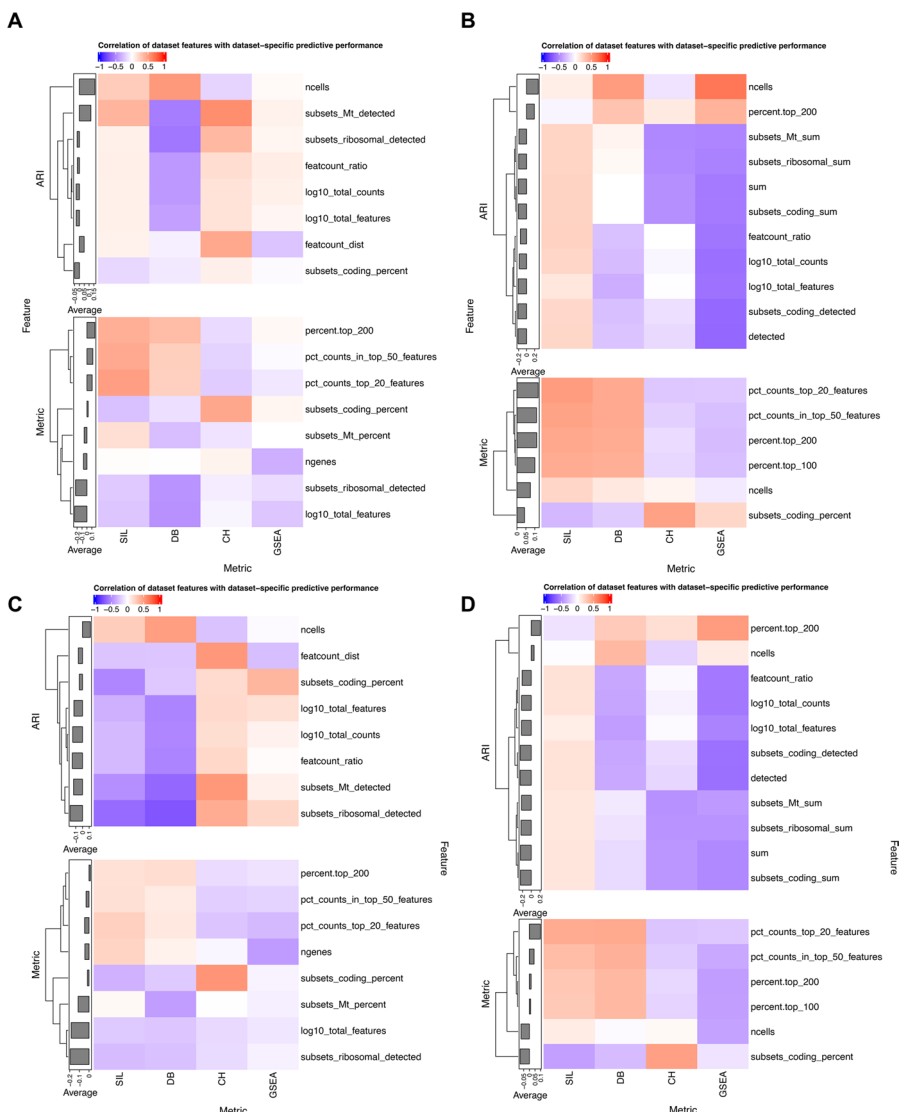

**Fig. 4** Correlation between predictive power and dataset-specific features for random forest models with dataset-pipeline interactions (**A**) and pipeline features only (**B**) and penalized linear regression models with interactions (**C**) and pipeline features only (**D**). In each panel, the predictive power is quantified by the correlation between model predictions and the ARI of the clusters and the ground truth (top, ARI) or by the correlation between model predictions and the true metric values (bottom, metric). Only features that are significantly associated with predictive power for at least one metric are shown

poor predictive performance on that dataset. This is important as, if a practitioner were to apply such models to a dataset with characteristics associated with strong predictive power, they are more likely to trust the resulting recommended pipeline, while in other cases they may wish to manually tune that pipeline. We identified such characteristics by correlating each dataset-specific feature with the metric-specific or ARI-specific predictive performance (which is itself the correlation between the predicted and actual metric values across pipelines or between the predicted values and the cluster overlap measured by ARI).

The results for a subset of features (see the "Methods" section) are shown for RF and LR in Fig. 4A, B, C and D respectively. This is evaluated for both pipeline-dataset interaction models and pipeline-only models on both metric-based and ARI performance measures. Notably, predicting SIL and DB has higher accuracy on datasets with a higher percentage of counts in the top 20/50/100/200 highly expressed genes per cell, a pattern weakly reversed in the prediction of CH and GSEA. On the other hand, being able to predict the CH metric for each dataset is highly correlated with the proportion of genes from the coding genome for both LR and RF models, a pattern that is not present for predicting the other metrics. For datasets with more cells, most models had better prediction performance for the pipelines that had better agreement with the ground truth. Importantly, associations between dataset characteristics and predictive performance were largely consistent (Additional file 2: Table S6) between LR and RF models, lending credence to the overall results. Together, these results provide a roadmap for interpreting which datasets may benefit from predictive modeling of pipeline performance.

Finally, for the random forest and linear regression models, we analyzed which features were most important in driving prediction values. For random forest, this is quantified as "IncNodePurity," which approximates the relevance of each feature at discriminating the outcome, and for penalized linear regression, we report the coefficient absolute value. For random forest (Additional file 1: Fig. S8), we found clustering resolution to be the most important feature for predicting 3/4 metrics, with filtering strategy, # dimensions for dimensionality reduction, and normalization strategy being the most important otherwise. In contrast, for linear regression, filtering strategy rather than clustering resolution was most important in 3/4 datasets (Additional file 1: Fig. S9). Interestingly, in this setting, interactions between dataset- and pipeline-specific features dominated as the most important features for all four metrics. These results suggest that particularly for random forest, clustering resolution is an important determinant of model performance, but other features and feature interactions are also crucial for prediction.

## Discussion

One limitation in this study is the relatively small number of scRNA-seq datasets used (86, albeit 288 pipeline observations from each), which necessitated parsimonious machine learning models (penalized linear regression, random forest) to not overfit. Since starting this work large-scale efforts such as CELLxGENE [27] have compiled > 1200 datasets with > 80 M cells total, raising the possibility that this benchmarking dataset may be expanded and more complex models such as deep learning may be applied. However, such datasets bring new challenges of scaling algorithms to many

cells. For example, when we further attempted to assess the performance of 288 pipelines on 6 datasets with > 100,000 cells, we found the implementation of scran normalization used was unable to handle these data sizes, resulting in pervasive missing data. Therefore, applying such approaches to larger scRNA-seq datasets will require careful consideration and selection of highly scalable algorithms.

A related limitation is the limited dataset-specific features to which we have access: these were either QC metrics or reduced-dimension gene expression features. Our attempts to incorporate additional dataset-specific metadata were made difficult by large amounts of missing metadata across studies or metadata unique to individual studies. Future efforts may benefit from further standardization of study metadata.

A further limitation concerns the use of prior cell type/cluster annotations as a measure of ground truth. While this has been utilized in existing benchmarking and method comparison papers, it has several drawbacks. Firstly, given there is no perfect clustering of data [28], these annotations may be subjective and vary between practitioners. Secondly, given that our benchmarking dataset covers a set of highly used pipelines, it may be that these were run by default to obtain the annotations, making the annotations biased towards certain pipelines.

In addition, here we considered only R-based pipelines, while there is an active and mature ecosystem of Python-based workflows for single-cell analysis as implemented through packages such as Scanpy [29]. This was motivated by both the already combinatorial number of R-based pipelines considered as well as existing tools for efficiently executing many possible R-based pipelines on a dataset [11]. While many Python single-cell analysis pipelines approximate R-based ones, recent work has thrown doubt on the extent to which the results are consistent between them [30]. Therefore, future work may wish to incorporate recommendations of Python-based pipelines in addition.

A further promising direction for future investigation is the incorporation of automated cell annotation methods as a possible metric and/or evaluation criteria. Multiple methods proposed by the community [31] take an annotated reference atlas and attempt to assign each cell in an unlabeled dataset to a previously annotated cell type based on gene expression similarity. While we have not applied such methods here due to the requirement to select an appropriate reference and annotation method, future machine learning algorithms may recommend pipelines based on agreement with automated cell type annotations or weight the deviation from automated annotation in their recommendation.

Finally, while the models we have used here frequently have predictive power on held out datasets that is significantly better than random, this is not equivalent to them having practically significant predictive power. Indeed, the average correlations between predicted and measured metric values range from 0.016 to 0.296 leaving much scope to improve with larger datasets and more powerful predictive models. However, given that there are currently no tools or studies to automatically recommend scRNA-seq analysis pipelines for a given dataset, we believe this is a useful starting point for further exploration and discussion in this field.

## Conclusions

In this study, we created a new dataset of four unsupervised clustering metrics applied to 288 analysis pipelines across 86 scRNA-seq datasets. By training supervised models to predict these metrics using pipeline and dataset-specific features, we demonstrated the exciting possibility of systematic recommendations of pipeline parameters for unseen datasets. Intriguingly, dataset-specific recommendations result in higher prediction accuracy when predicting the metrics themselves but not necessarily when considering whether predictions align with prior clustering results. To stimulate methods development in the community in this area, we release the resulting dataset containing the 4 clustering metrics from 24,768 unique clustering outputs as the *Single Cell pIpeline PredIctiOn (SCIPIO-86)* dataset, along with the pipeline- and dataset-specific features required for building predictive models.

## Methods

### Dataset collation

The scRNA-seq datasets used in this study were sourced from the European Bioinformatics Institute's Single Cell Expression Atlas [16]. The selected datasets were the 86 datasets that were (i) human and (ii) contained fewer than 100,000 cells as of May 2021. The upper bound on the number of cells considered was due to memory and time constraints when applying many pipelines. For each dataset, a large number of dataset-specific features were computed using the scuttle package addPerCellQC function [32] (Additional file 2: Table S2). The code for all analyses in this paper can be found at github.com/camlab-bioml/beyond_benchmarking_analyses.

### scRNA-seq pipeline construction

In total, 288 different clustering pipelines were run on each dataset using the pipeComp framework [11]. These pipelines consisted of different (i) filtering, (ii) normalization, (iii) dimensionality reduction, and (iv) clustering methods and/or parameters (Additional file 2: Table S7). All datasets were fed into the clustering pipelines as raw counts with per-cell quality control metrics computed using scuttle's addPerCellQC as above as the quality control metrics were required for preprocessing steps such as filtering.

While the different steps of the pipelines are well described in the pipeComp paper, we briefly describe them below:

1. *Filtering.* The *default* method excludes cells that meet at least two of the following criteria: *log10_total_counts* less than 5 median absolute deviations (MADs) or greater than 2.5 MADs, *log10_total_features* less than 5 MADs or greater than 2.5 MADs, *pct_counts_in_top_20_features* greater than or less than 5 MADs, *featcount_dist* greater than or less than 5 MADs, *pct_counts_Mt* greater than 2.5 MADs and greater than 0.08. The *stringent* filtering method uses the same thresholds as *default*, but a cell only needs to meet one of the above criteria before being discarded. *Lenient* filtering excludes cells with metrics greater than 5 MADs on any two quality control metrics, except for *pct_counts_Mt* where cells with a percentage of mitochondrial counts greater than 3 MADs or greater than 0.08 were excluded.

2. *Normalization.* Three different normalization methods were computed. These were scran's pooling-based normalization [18], sctransform's variance-stabilizing normalization [19], and seurat's log-normalization [17], all with default parameters. After normalization, feature selection was performed using Seurat's FindVariableFeatures function in the "vst" setting, and the top 2000 most variable features were kept.

3. *Dimensionality reduction.* The dimensionality reduction step was performed using Seurat's PCA method and the dimension parameters considered were 10, 15, 20, and 30.

4. *Clustering* was performed using Seurat, with resolutions 0.1, 0.2, 0.3, 0.5, 0.8, 1.0, 1.2, and 2.0. Since the number of clusters should reflect the number of true sub-populations in each dataset, clustering was performed with many different resolutions as the true number of subpopulations is not known a priori.

### Metrics of pipeline success

The results from the clustering pipelines were evaluated using the unsupervised cluster purity metrics from sklearn [33]: Calinski-Harabasz index (CH), Davies-Bouldin index (DB), and silhouette coefficient (SIL). These were applied to the $\log(x+1)$ count matrix filtered to contain the same cells as the clustering output for that pipeline (after e.g., filtering) on the 500 most highly variable genes. CH measures the ratio of the sums of between-cluster dispersion and of within-cluster dispersion for all clusters. A higher CH value indicates that clusters are dense and well-separated, which generally means a better clustering output. DB measures the average similarity between each cluster and its most similar one, with similarity defined as the ratio between the sum of *cluster diameters* (mean distance between each point in a cluster and its centroid) and the distance between cluster centroids. Unlike CH, a higher value of DB indicates a worse clustering performance, with values closer to 0 representing better, well-separated clustering. SIL is a measure of how cohesive and separated clusters are, quantified as the mean score across samples. For each sample, SIL computes the (normalized) difference between the mean distance from that sample to points in the same cluster and mean distance to points in the nearest cluster. SIL is bounded between $-1$ and 1, and, similarly to CH, higher scores indicate better clustering. In contrast to SIL and CH, a lower DB indicates a more favorable clustering, so for the remaining analyses, we multiplied the DB values by $-1$ for the direction of change in clustering performance to be consistent across metrics. After the above metrics were computed, they were scaled to have mean 0 and variance 1 within each dataset since their values are influenced by dataset-specific biological and technical factors and are thus not comparable between datasets.

### Adjusting metrics for number of clusters identified

The cluster purity metrics were corrected to remove the relationship between the metric values and the number of clusters found by each pipeline. For each dataset, we fit a loess model using the loess function in R with the formula metric ~ number of clusters. We

then extracted the residuals from each of the loess models and used the residuals as our corrected metric values.

### GSEA clustering metric

Aside from cluster purity metrics, Gene Set Enrichment Analysis (GSEA) [21] was used as an additional metric. For each of the clustering outputs, findMarkers from the scran R package [34] was used to identify the marker genes for each cluster. The log fold change for the marker genes of each cluster were then mapped to gene symbols and GSEA was computed using the Gene Ontology Biological Processes, Cellular Component, and Molecular Function gene sets, as well as the Human Phenotype Ontology gene sets [21] using the fgsea package [35]. These gene sets were subsetted to only include the sets with more than 10 and fewer than 500 genes (but no FDR filtering was performed). After an absolute normalized enrichment score (NES) was computed for each cluster, they were averaged across all clusters to yield one *GSEA metric* for each clustering output. The GSEA metrics were then scaled to have mean 0 and variance 1 within each dataset in line with the other metrics.

### Comparison to expert-derived labels

We were able to compute the adjusted Rand index (ARI) using sklearn on 16 of the 86 datasets as they included cell type labels. ARI is a standard measure of clustering performance that has been used in many other benchmarking studies when ground truth is available. The Rand index is a measure of the similarity between two different clusterings, and the ARI adjusts the Rand index for chance overlap due to increasing cluster numbers. The ARI reaches a maximum value of 1 when the two clusterings are the same and is close to 0 when the cells are labeled randomly.

### Handling missing data

We found that some pipeline and dataset combinations yielded clustering outputs consisting of a singular cluster, leading to missing values in the unsupervised metrics as their computation requires at least two distinct clusters. We imputed the missing values using the median scaled metric value of each dataset. For the cluster purity metrics, this was done after correcting the metrics for the number of clusters.

### Predicting pipeline performance using pipeline parameters and dataset characteristics

For each of the four metrics, random forest and linear regression models were used to predict the metric, given pipeline parameters and dataset characteristics as inputs. Each observation given to the model represents a unique pipeline and dataset combination. The input data consisted of 45 features. Of this 45, 20 features were the first 20 principal components of gene expression, four were pipeline parameters, and the remaining 21 were dataset quality control metrics. The quality control metrics were per-cell metrics such as the number of genes detected and percentage of mitochondrial counts (see Additional file 2: Table S1). For each dataset, we used the median of each per-cell metric as the input to our machine learning models.

Principal component analysis (PCA) was used to create the additional 20 gene expression features mentioned above. For each scRNA-seq dataset, we summarized it with a

single vector by taking the mean expression of each gene's raw counts. We then collated these vectors into a dataset by gene matrix. Notably, this is conceptually different from the standard PCA run on the gene by cell expression matrices in the typical scRNA-seq data analysis workflow. PCA on the dataset by gene matrix will capture the directions of variation in the mean gene expression across the 86 datasets. Using the same train-test split as below, we first fit a probabilistic PCA model [36] on the training set and then used the fitted model to transform the test set. The scores were then extracted from both the train and test sets and used as input features for the machine learning models.

The models were trained using a 70/30 dataset-aware train-test split where all pipeline results for a given dataset must be included in either the train or test set. Sixty-one of the 86 datasets (70%) were placed in the training set, and the remaining 25 (30%) were placed in the test set. This was done to ensure that our models would be able to predict pipeline performance on completely unseen datasets, which is a more realistic scenario as practitioners generally do not have access to benchmarking results for their specific dataset. Within the 25 datasets in the test set, we included the 16 datasets with cell type labels. This allowed us to evaluate our models' test predictions against ground truth performance.

Since we modeled each of the four metrics as a function of dataset-specific characteristics and pipeline parameters, we scaled all numeric values before giving them as inputs to the models so that the predictions would not be dominated by inputs on extreme scales. We computed the means and standard deviations of each of the numeric features on the train set and used them to center both the train set and test set.

### Hyperparameter tuning

Tenfold dataset-aware cross-validation (CV) on the train set was used to tune model hyperparameters using the R package caret [37], meaning for each CV iteration, results from one dataset would appear in one fold only. We opted for a dataset-aware CV process as not being dataset-aware would give test or cross-validation performance equivalent to already having access to some ground truth values for a given dataset—in other words, the datasets would not be "unseen." The number of trees (*ntree*) and number of variables tried at each split (*mtry*) were tuned using grid search for the random forest models, which were fitted using the randomForest R package. Values of 100, 300, and 500 were considered for *ntree* and values of 1 to the total number of features (45 and 4 for the dataset-pipeline interactions and pipelines-only models respectively) for *mtry*. The linear regression models were fitted using the glmnet R package [38]. The hyperparameters $\alpha$ and $\lambda$, which control the gap between lasso and ridge regression and the strength of the penalty term respectively, were tuned using grid search as well. Three values were tried for both $\alpha$ and $\lambda$ using caret defaults. For both the random forest and penalized linear regression models, the final models were selected based on the hyperparameters that minimized the cross-validated root mean squared error.

### Evaluating model performance

We evaluated the performance of the models by comparing their predictions with the corresponding calculated corrected metrics. For each dataset in the test set, we computed the Pearson correlation between the model's predictions and the observed metric

value. We then computed a one-sided Wilcoxon rank-sum test against 0 for all the correlations in the test set for each model and used Benjamini–Hochberg multiple test correction. This allowed us to evaluate whether the models were able to predict on the test set with a significantly positive correlation with the observed metric values compared to random. We repeated this process on the 16 labeled datasets but replaced the observed metrics with the ARI between the clustering returned by a given pipeline and the existing labels to measure the agreement between our predictions and previous cell type annotations.

### Quantifying determinants of dataset-specific prediction performance

Since the correlation between the predictions of our models and the observed metric varied between datasets, we examined which features were indicative of pipeline predictive performance. We quantified predictive performance using the Pearson correlation between the model's predictions and the observed metric value for each dataset. We then computed the Pearson correlation between the predictive performance and dataset-specific features across datasets, giving us one correlation for each feature. Finally, we performed a two-sided association test for these correlations to determine whether they were significantly different from 0. On the 16 datasets with ground truth labels available, we also quantified predictive performance as the correlation between the models' predictions and the ARI. Using this measure of predictive performance, we repeated the above process to determine which dataset-specific features impact predictive ability. In the heatmaps in Fig. 4, the dataset-features have been filtered to only include those that have a correlation significantly different to 0 in at least one of the four models (CH, DB, SIL, GSEA).

### Supplementary Information

> Additional file 1. All supplementary figures for this study.
>
> Additional file 2. All supplementary tables for this study.
>
> Additional file 3. A copy of the sample-level metadata for the 86 datasets used in this study.
>
> Additional file 4. Review history.

**Acknowledgements**
We thank Hassaan Maan and Chengxin Yu along with two anonymous reviewers for their feedback on this manuscript.

**Peer review information**

**Review history**
The review history is available as Additional file 4.

**Authors' contributions**
Concept: KRC, CF, AS. Data analysis: CF. Paper writing: KRC, CF, AS.

**Funding**
This work is funded by NSERC Discovery Grant (RGPIN-2020–04083) to KRC, a CIHR project grant (PJT-175270) to KRC, and KRC acknowledges support from the Canada Research Chairs program and the Canadian Foundation for Innovation. AS acknowledges support from the Hold'em for Life Oncology Fellowship, the Canadian Statistical Sciences Institute, and the Vector Institute. CF acknowledges support from the Undergraduate Research Opportunities fund (UROP) in the Department of Molecular Genetics, University of Toronto.

**Availability of data and materials**

Code to reproduce all results and figures is available at https://github.com/camlab-bioml/beyond_benchmarking_analy ses [39] and at https://zenodo.org/records/11402899 [40]. All data and code are released under a GNU General Public License.

The scRNA-seq datasets used in this study are summarized in Additional file 2: Table S1 and are all publicly available through the EBI Single Cell Expression Atlas [16]. Pipeline performance metrics on these datasets are curated in the SCIPIO-86 dataset available at https://zenodo.org/records/11403435 [41].

## Declarations

**Ethics approval and consent to participate**
Not applicable.

**Consent for publication**
Not applicable.

**Competing interests**
The authors declare no competing interests.

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
