## [Additional file 4. Review history. · Genome Biology]

Review History

First round of review

Reviewer 1

Are you able to assess all statistics in the manuscript, including the appropriateness of statistical tests used? Yes, and I have assessed the statistics in my report.

Comments to author:

Fang et al present a manuscript describing the generation of a dataset comprising 86 single-cell RNA-seq data sets run on 288 processing pipelines, with the goal of trying to predict the "best" pipeline for a particular data set. This is well motivated, drawing parallels with AutoML for the selection of workflow parameters, and reduce the workload burden of analysts. The goal is also to allow users to select the optimal pipeline for their unique dataset, based on the authors analysis. This is a new area that goes beyond the standard approach of benchmarking methods on a limited number of data sets, and the authors should be commended for taking this next step. While they provide a convincing proof of principle that it is possible to predict scRNA-seq pipeline performance, my enthusiasm is slightly tempered by the lack of useful guidelines or recommendations that users could consider. Below are my specific comments and queries that I hope could be used to clarify the manuscript for a broad user-base.

The authors used datasets with <100k cells - which is motivated by consideration on compute resources. Given the number of cells appears to be strongly correlated with model performance and therefore prediction performance, is there a study size after which it no longer matters? For instance, if the authors were to evaluate some pipelines on a dataset(s) on the scale of $\leq 10^6$ cells, does this distinction disappear, and do other features become important instead? This could be very informative for large atlas and patient-cohort scale studies which typically contain 500,000+ cells.

Focus is on R-based pipelines. What about Python-based ones, e.g. Scanpy?

It seems that while the RF and regression predictions are positively correlated with the compute metric values, the correlation coefficients are still quite modest (median <0.5). While clearly there is some non-randomness as the correlations are largely > 0 , this illustrates that there is still a lot of information not captured by the RF/LR models. For the ARI, this is even less convincing, modulo the caveats of relying on the provided study annotations as a ground truth. Could the authors expand on this a bit more than they do in the discussion? Perhaps by considering other factors of pipelines or datasets, such as technology, tissue of origin, cell type diversity, etc that could be considered to (a) improve prediction performance, and (b) guide users in selecting the best pipeline for their data set?

Given the PCA captures most of the variation in the gene expression data, and these are used to build NN-graphs for performing clustering on, do the PCs tend to drive the increased performance in the dataset-pipeline ML models? If so, this might be tautological as the ML model is essentially being fed some of the same information as each pipeline. Could the authors please comment on this?

For the GSEA-based evaluation, were all gene sets with >10 and <500 genes considered, or was there a filtering on the enrichment of each, e.g. 10% FDR? For example, if there are say 100 gene sets and half are enriched in one cluster but only one in another cluster, is this a fair comparison as a metric of clustering performance? This may be driven by clustering resolution in a dataset-dependent manner.

In Figure 2A there appears to be roughly 2 groups of datasets where GSEA score is (1) positively correlated with clustering resolution and (2) where it is negatively correlated. Is there some common feature of these data sets driving this stratification?

In the results section, please give a brief description of the cluster purity metrics, what they show and why they illustrate different aspects of clustering pipeline performance. For instance, why the need for 3 metrics that internally evaluate clustering performance? Particularly given that CH and Silhouette appear correlated, and anti-correlated with DB?

In supplementary Figure 3 - despite the lack of consensus over data sets, are the highest-ranking pipelines on one metric consistent between metrics?

Supplementary Figure 4 is very small and hard to read - this makes it difficult to evaluate the ground truth labels vs predictions. Does the best-performing pipeline that has the highest ARI use processing steps that closely match those of the original study? Is that the explanation for the higher performance?

Reviewer 2

Are you able to assess all statistics in the manuscript, including the appropriateness of statistical tests used? Yes, and I have assessed the statistics in my report.

Comments to author:

The submitted manuscript by Fang, Selega, and Campbell explores the possibility of model-based approaches to predict optimal processing pipelines for scRNA-seq data sets.

I think field will benefit from exploration in this direction as datasets continually grow in scale and complexity. I support the publication of this work, but have some comments and think some revisions could help to give a better sense of how the output of an 'optimal' pipeline is better than blindly running a 'standard' pipeline.

Comment 1. Overall, I felt that it's not delivering a sense of how this optimization is leading to meaningfully improved results. Rather than just a comparison against a randomly selected pipeline that may not be overly representative (eg. I'd argue clustering resolutions >0.5 are rarely practical), are there demonstrable cases where the optimal pipeline is notably better than blindly putting the data through Seurat's generic "Getting Started" workflow?

Comment 2. Is there a way to get a sense of which pipeline parameters are having the greatest impact? My worry is that this is largely driven by clustering resolution, which if true, could be better handled on a per-sample basis with the various approaches to optimize clustering

resolution. It's not the end of the world if that's the case because I think this approach could be developed to a point where we're not just talking about clustering, but it's worth knowing.

Comment 3. I appreciate the discussion about the limitations of using user-defined cell type annotations as ground truth. As mentioned, particularly the issue that cell types in these studies are very likely derived from manual annotation of clusters, which may result in the model simply trying to match the original pipeline. Or the original authors annotated at some granularity that may/may not be relevant (eg. Supp Fig 4's labels seem coarse given the number of clusters with label 1).

It would be interesting to instead test on a couple datasets that were annotated (previously or by you) with an automated method that operates on each cell based on its expression profile.

Comment 4. Since clustering resolution is regressed out of the metrics to produce the corrected metrics for downstream steps, should Figure 2A show the corrected heatmap rather than the uncorrected?

Comment 5. I understand that a decision must be made to stop data collection at some point and would not suggest you need to expand the collection. But the discussion (lines 285-286) gives the impression that larger models are now limited by data availability. It may be nice to recognize recent developments like CELLxGENE, which currently has a standardized collection of 1271 datasets and >80M cells.

Comment 6. The various dataset features are largely technical and related to quality. As a discussion point, could there be opportunities incorporating additional dataset features that may better reflect characteristics of the data? Eg. features from the count matrix, like variance. Or maybe an extension of that—what if every dataset was put through some generic pipeline, several metrics get calculated (even including the ones you're optimizing for), and then the model could predict what changes to the pipeline could improve it.

Comment 7. Can you confirm if FindVariableFeatures() was run following SCTransform? Typically the SCTransform() function uses residual variance from the model to define variable genes.

Reply to reviewers – Fang et al. 2024

We thank the reviewers for the time taken to review our manuscript and their comments and suggestions, to which we reply below. Our replies are in **blue** and updates to the manuscript in **red**. We have also re-formatted the manuscript as requested and made some minor aesthetic changes to Fig 1A & 1B.

Reviewer #1

1. The authors used datasets with <100k cells - which is motivated by consideration on compute resources. Given the number of cells appears to be strongly correlated with model performance and therefore prediction performance, is there a study size after which it no longer matters? For instance, if the authors were to evaluate some pipelines on a dataset(s) on the scale of $\leq 10^6$ cells, does this distinction disappear, and do other features become important instead? This could be very informative for large atlas and patient-cohort scale studies which typically contain 500,000+ cells.

We appreciate the reviewer's point and the fact that many single cell datasets of interest are larger scale atlases with >500k cells.

To address this, we retrieved the 16 datasets with >100k cells we had initially filtered out as part of our workflow and attempted to run all 288 pipelines on each. Here we faced two challenges: (i) we could not run the scran normalization (as implemented in pipeComp) on any dataset as even our large memory compute nodes (512GB) didn't have sufficient RAM, and (ii) in the 6 weeks given for revisions we were only able to run these 192 pipelines (all those without scran) across 6/16 datasets (as listed in **Supplementary Table 4**), giving 4608 unique values across the four metrics.

For each of the 6 large datasets, we compared the correlation between the predicted and actual metric values across the four metrics for the two machine learning models, and contrasted this with the equivalent values for the existing 25 test sets we considered in the paper (that have <100k cells):

It can be seen for random forest (left) and penalized linear regression (right) the results are roughly comparable between the two classes of datasets (or even somewhat improved). Intriguingly, for the CH metric, both the random forest and penalized linear regression models show a meaningfully improved correlation on the >100k cell datasets, though we caution with only 6 datasets we are likely underpowered to make a definitive statement about improved performance and this could also be caused by removing scran as a possible normalization procedure.

With respect to the reviewer’s original question about which features become more or less important in the >100k cell regime, this analysis relies on correlating dataset specific features with dataset specific performance. We are therefore underpowered with N=6 datasets (and would still likely be with N=16) to make any definitive statements, compared to the originally presented set of datasets with <100k cells for which we had N=88, which is partially driven by scarcity of both very large datasets and compute capacity. In addition, with N=6 we would not be able to re-train the model to assess feature importance. However, given the prediction performance is broadly comparable in the >100k cell regime above (despite the small sample size), we would expect similar features to remain important. To address this, we have added the following to the main text:

Next, we investigated if the predictive performance of these models differs in atlas-scale datasets containing >100,000 cells. We processed an additional 6 datasets (**Supplementary Table 4**) with the same pipelines as before (except for those using scran normalization that gave an out of memory error) and applied the trained predictive models from above to them. Correlating the predicted and actual metric values shows broadly comparable or improved performance (**Supplementary Fig. 4**), implying such predictive models will scale to larger and more complex datasets.

2. Focus is on R-based pipelines. What about Python-based ones, e.g. Scanpy?

Our motivation to focus on R-based pipelines was motivated by two factors: (i) even within R only pipelines there is a combinatorial number of possibilities (here we consider

288), so incorporating Python pipelines in addition would have expanded the pipeline space (and the required compute) to an infeasible level, (ii) previous work has provided elegant bioinformatics tooling for evaluating multiple R based pipelines simultaneously (<https://genomebiology.biomedcentral.com/articles/10.1186/s13059-020-02136-7>), reducing the required tooling.

However, we acknowledge this is a limitation of the results presented here, so have added the following paragraph to the discussion:

In addition, here we considered only R-based pipelines, while there is an active and mature ecosystem of Python-based workflows for single-cell analysis as implemented through packages such as Scanpy [25]. This was motivated by both the already combinatorial number of R-based pipelines considered as well as existing tools for efficiently executing many possible R-based pipelines on a dataset [11]. While many Python single-cell analysis pipelines approximate R-based ones, recent work has thrown doubt on the extent to which the results are consistent between them [26]. Therefore, future work may wish to incorporate recommendations of Python-based pipelines in addition.

3. It seems that while the RF and regression predictions are positively correlated with the compute metric values, the correlation coefficients are still quite modest (median <0.5). While clearly there is some non-randomness as the correlations are largely > 0 , this illustrates that there is still a lot of information not captured by the RF/LR models. For the ARI, this is even less convincing, modulo the caveats of relying on the provided study annotations as a ground truth. Could the authors expand on this a bit more than they do in the discussion? Perhaps by considering other factors of pipelines or datasets, such as technology, tissue of origin, cell type diversity, etc that could be considered to (a) improve prediction performance, and (b) guide users in selecting the best pipeline for their data set?

This is an excellent suggestion and something we initially overlooked. To attempt to answer this, we turned to the study metadata as captured on EBI's SingleCellExperiment atlas. For additional metadata to be useful for performance prediction, it needs to be (i) not missing in the majority of datasets, (ii) not the same across all datasets, and (iii) if a categorical variable, only take on a relatively small number of possibilities given the size of the training dataset we have. To further exemplify the last point, we have 86 datasets, so if there's a categorical metadata column with e.g. 60 different values then the model will only see each on average once and therefore not generalize. Examining the metadata (which we attach as **Supplementary Data Sheet 1**), there was no column with continuous data with a low ($<50\%$) missingness rate. Therefore, for the remaining categorical columns, we plotted the number of unique values against the number of missing values:

Number of unique entries vs. number of missing entries per metadata field

As can be seen there are relatively few columns with a low missing % and a low (but >1) number of unique values. Therefore, we derived 3 additional dataset features that could be assigned to each dataset with a low missing %:

1. End bias (5', 3', none)
2. Had the cells been FACS sorted? (yes, no)
3. Did the cells originate from blood? (yes, no)

We retrained our random forest and penalized linear regression models conditioning on these additional extra data features and reported the test set performances:

Compared to the original values not incorporating these features, the results are broadly comparable (the p-values for random forest are slightly higher than before, and slightly lower for penalized linear regression, but it feels hard to argue there's a meaningful difference). Overall, this illustrates the difficulty of using user provided dataset metadata beyond what can be directly computed from the expression profiles, given its high missingness rate and non-standardized entries. However, we believe this is an important analysis, so therefore have added the above as **Supplementary Fig. 7** referred to in the main text as follows:

Finally, we considered if additional metadata provided by each original study's authors would be of benefit for predicting dataset-specific pipeline performance. Given the large proportion of missing entries in each study's metadata (**Supplementary Data**), we focused on three that were near-universally available for all datasets: (i) the end-bias of the sequencing protocol (5', 3', or none), (ii) whether the cells had been FACS sorted (yes or no), and (iii) whether the cells originated from blood (yes or no). After retraining the machine learning models to predict dataset-specific pipeline performance incorporating these features, we found largely equivalent model performance (**Supplementary Fig. 7**).

And have expanded the discussion to incorporate the above:

A related limitation is the limited dataset-specific features to which we have access: these were either QC metrics or reduced-dimension gene expression features. Our attempts to incorporate additional dataset-specific metadata were made difficult by large amounts of missing metadata across studies or metadata unique to individual studies. Future efforts may benefit from further standardization of study metadata.

4. Given the PCA captures most of the variation in the gene expression data, and these are used to build NN-graphs for performing clustering on, do the PCs tend to drive the increased performance in the dataset-pipeline ML models? If so, this might be tautological as the ML model is essentially being fed some of the same information as each pipeline. Could the authors please comment on this?

The PCs of gene expression used as input features for our model differ from those used for clustering in two important ways. Firstly, the input feature PCs are computed on mean gene expression across the training datasets, rather than the single-cell counts matrix for each individual dataset. Secondly, the PCA rotations used for prediction are fitted using the train data only. The features (i.e. components) used in the test set are the test set expression projected using the train rotations to avoid train-test leakage. Consequently, the PCA features used for clustering the test datasets are likely different (or rather, not necessarily the same) as those used for pipeline prediction, and so given there is no train-test leakage the performance isn't tautological (if we have interpreted the reviewer's comment correctly). This was described in the original manuscript as follows, though we would be happy to clarify this if needed:

"Using the same train-test split as below, we first fit a probabilistic PCA model [32] on the training set, and then used the fitted model to transform the test set. The scores were then extracted from both the train and test sets and used as input features for the machine learning models."

5. For the GSEA-based evaluation, were all gene sets with >10 and <500 genes considered, or was there a filtering on the enrichment of each, e.g. 10% FDR? For example, if there are say 100 gene sets and half are enriched in one cluster but only one in another cluster, is this a fair comparison as a metric of clustering performance? This may be driven by clustering resolution in a dataset-dependent manner.

There was no filtering based on FDR, so all gene sets with >10 and <500 genes were considered. We have clarified this in the Methods:

“These gene sets were subsetted to only include the sets with more than 10 and fewer than 500 genes (but no FDR filtering was performed).”

6. In Figure 2A there appears to be roughly 2 groups of datasets where GSEA score is (1) positively correlated with clustering resolution and (2) where it is negatively correlated. Is there some common feature of these data sets driving this stratification?

To answer this, we split each dataset into the two groups based on positive/negative correlation with GSEA score and compared feature values in each group:

The largest determinant appears to be the number of cells, with the positively correlated group having more cells on average. However, this association is not absolute: some studies with a high number of cells are in the negatively correlated group, and some studies with very few cells are in the positively correlated group. Indeed, differences between the two groups are likely somewhat explained by this resulting in more reads per cell in the negative group, with this group exhibiting a higher gene detection rate and higher total counts per cell.

7. In the results section, please give a brief description of the cluster purity metrics, what they show and why they illustrate different aspects of clustering pipeline performance. For instance, why the need for 3 metrics that internally evaluate clustering performance? Particularly given than CH and Silhouette appear correlated, and anti-correlated with DB?

We have added the following to the main text giving a brief description of each metric:

CH measures the ratio of between-cluster dispersion to within-cluster dispersion, favoring well-separated, dense clusters. DB measures cluster similarity by comparing each cluster to its most similar one, attributing good scores to distinct, well-separated clusters. SIL measures how well each data point fits into its assigned cluster, with higher scores signifying consistent clusters (a point is closer to members of its own cluster) and negative scores suggesting misclassification.

We chose to include three metrics for several reasons. Firstly, it was not certain (though is perhaps expected) they would necessarily be correlated across datasets and the variability with which they can be predicted demonstrates some level of independence. For example, optimal pipelines predicted using DB typically have higher ARI with ground truth annotations compared to the other metrics, while the models can consistently predict CH better than the other metrics. Secondly, if we didn't include all three we would have to pick one, and it is not obvious (to us) *a priori* which should be included beyond the others given all have successfully been used in benchmarking papers. To summarize this, we have added (immediately below the above):

We opted for these three metrics as it was not *a priori* obvious if they would be correlated nor which one to favor above the others, and each may be computed for any clustering result in the absence of additional information such as cell labels.

8. In supplementary Figure 3 - despite the lack of consensus over data sets, are the highest-ranking pipelines on one metric consistent between metrics?

We found this was not the case:

Pipeline Step	Best CH pipeline on average	Best DB pipeline on average	Best SIL pipeline on average	Best GSEA pipeline on average
Filtering	Default	Stringent	Stringent	Default
Normalization	Scran normalization	Seurat	Seurat	Scran normalization
PCs used for clustering	30	30	30	15
Clustering resolution	0.1	0.1	0.1	1.2

We have added this as Supplementary Table 2 referenced in the updated manuscript:

“Using corrected metrics to measure pipeline success, we found no single pipeline performed best across all datasets (**Supplementary Fig. 3**) and the best pipeline on average was not the same across each metric (**Supplementary Table 2**).”

9. Supplementary Figure 4 is very small and hard to read - this makes it difficult to evaluate the ground truth labels vs predictions. Does the best-performing pipeline that has the highest ARI use processing steps that closely match those of the original study? Is that the explanation for the higher performance?

We apologize for the size of S. Fig. 4 – we have enlarged it to improve readability in the revised manuscript. To answer the second part, it appears the best predicted pipeline was highly similar to that used by the original authors, which given that neither the authors’ pipeline nor labels were supplied to the method lends credibility to the overall approach:

Pipeline step	Original authors’ parameters	Best predicted pipeline’s parameters
Doublet detection	None	None
Filtering	Not described in paper	Stringent filtering
Normalization	Seurat normalization	Seurat normalization
Feature selection	Not described in paper	sel.vst
Dimensionality reduction	Seurat PCA, number of principal components not described	Seurat PCA with 50 principal components
Clustering	Seurat clustering, resolution=0.3	Seurat clustering, resolution=0.2

Reviewer #2

Comment 1. Overall, I felt that it's not delivering a sense of how this optimization is leading to meaningfully improved results. Rather than just a comparison against a randomly selected pipeline that may not be overly representative (eg. I'd argue clustering resolutions >0.5 are rarely practical), are there demonstrable cases where the optimal pipeline is notably better than blindly putting the data through Seurat's generic "Getting Started" workflow?

To illustrate this, we have provided examples on two datasets comparing the Seurat default pipeline to the best predicted pipeline and the author’s original labels. E-MTAB-9221 is a scRNA-seq dataset of whole blood (with RBC depletion) from patients with Covid-19 and controls:

It can be seen the default Seurat workflow effectively overclusters the data, splitting multiple cell types (e.g. neutrophils, monocytes, B cells) into multiple subclusters, while the best predicted pipeline via our investigation does not. Note that the same set of cells is not necessarily present in each comparison as different filtering criteria are used.

However, it could be argued from the above that the major benefit from the recommendation is closer matching of the resolution/number of clusters. Therefore, we selected a further dataset (E-MTAB-8410, scRNA-seq of colorectal tumours & adjacent tissue) to compare as before. Here, the default Seurat pipeline finds 26 clusters, much closer to the 41 cell types annotated by the authors compared to 16 as recommended by our pipeline:

Despite this, numerous examples of Seurat default pipeline splitting up author labelled ground truth annotations are present *despite having fewer clusters specified than the author labels*, while the best predicted pipeline does so to a lesser extent:

We believe these demonstrations are important for readers getting an intuitive understanding of the differences and therefore have summarized the above as a supplementary figure and added the following to the main text:

Next, we examined the qualitative differences between the best predicted pipeline and the Seurat version 5.0.3 default pipeline on two representative datasets from our test data for which author ground truth annotations were available. First, we examined a scRNA-seq dataset of whole blood from patients with Covid-19 and controls [24]. We found the best predicted pipeline learned 12 cell clusters that matched closer to the 6 cell types annotated by the authors compared with the 18 found by the Seurat default pipeline (**Supplementary Fig. 6A**). However, this may be simply due to the predicted pipeline learning fewer clusters overall. Therefore, we examined an additional dataset of scRNA-seq of colorectal tumours [25] where the Seurat default pipeline learns a number of clusters (n=26) much closer to the annotated number (n=40) than that returned by the best predicted pipeline (n=16) (**Supplementary Fig. 6B**). Despite this, the default Seurat pipeline splits several annotated clusters into multiple subclusters, while the best predicted pipeline does not or does to a far lesser extent (**Supplementary Fig. 6C**).

Comment 2. Is there a way to get a sense of which pipeline parameters are having the greatest impact? My worry is that this is largely driven by clustering resolution, which if true, could be better handled on a per-sample basis with the various approaches to optimize clustering resolution. It's not the end of the world if that's the case because I think this approach could be developed to a point where we're not just talking about clustering, but it's worth knowing.

This is an important point that we initially overlooked and thank the reviewer for bringing it to our attention. To address this, for each model (random forest, penalized linear regression) and the four metrics we predict, we compute a feature importance measure. For random forest, this is the “IncNodePurity”, which approximates the relevance of each feature at discriminating the outcome, and for penalized linear regression we report the coefficient absolute value, which (approximately) represents feature importance given the input features were scaled. This gives the following top feature importance values per model/metric:

Variable Importance for CH RF

Variable Importance for DB RF

Variable Importance for SIL RF

Variable Importance for GSEA RF

Variable Importance for SIL LR

Variable Importance for GSEA LR

Variable Importance for CH LR

Variable Importance for DB LR

There are several interesting aspects to comment on here. For random forest, the reviewer's intuition is approximately correct in that resolution ("res") is the most important feature in 3/4 metrics. However, for CH, filtering and normalization strategy are ranked higher, and for the other metrics, many other features (notably, filtering strategy, # dimensions for dimensionality reduction, and normalization strategy) have large feature importance values also. In contrast, for the penalized linear regression model, the resolution no longer dominates as an important feature, with filtering strategy typically being represented among the most important features, and interestingly interactions between dataset and pipeline features dominating (which is intuitively perhaps expected).

Our conclusion would therefore be that the reviewer is correct in the sense that clustering resolution is an important feature but that the results aren't solely driven by clustering resolution. We have added the above two figures as supplementary figures and the following to the main text:

Finally, for the random forest and linear regression models we analyzed which features were most important in driving prediction values. For random forest, this is quantified as "IncNodePurity", which approximates the relevance of each feature at discriminating the outcome, and for penalized linear regression we report the coefficient absolute value. For random forest (**Supplementary Fig. 8**), we found clustering resolution to be the most important feature for predicting 3/4 metrics, with filtering strategy, # dimensions for dimensionality reduction, and normalization strategy being the most important otherwise. In contrast, for linear regression, filtering strategy rather than clustering resolution was most important in 3/4 datasets (**Supplementary Fig. 9**). Interestingly, in this setting interactions between dataset and pipeline specific features dominated as the most important features for all four metrics. These results suggest that particularly for random forest, clustering resolution is an important determinant of model performance but other features and feature interactions are also crucial for prediction.

Comment 3. I appreciate the discussion about the limitations of using user-defined cell type annotations as ground truth. As mentioned, particularly the issue that cell types in these studies are very likely derived from manual annotation of clusters, which may result in the model simply trying to match the original pipeline. Or the original authors annotated at some granularity that may/may not be relevant (eg. Supp Fig 4's labels seem coarse given the number of clusters with label 1.

It would be interesting to instead test on a couple datasets that were annotated (previously or by you) with an automated method that operates on each cell based on its expression profile.

We thank the reviewer for this suggestion and think the idea of using automated cell type annotations as an additional metric is an interesting direction. To test this idea, we used one of the datasets already included in the study (E-GEOD-114530), which is scRNA-seq of human fetal kidneys, given the relative ease of specifying a reference

atlas, for which we used the HCA human fetal kidney atlas (<https://www.science.org/doi/10.1126/science.aat5031>). We annotated each cell in E-GEOD-114530 using the cell annotation reference atlas using SingleR v. 2.4.1. For the clusters returned by each pipeline we then computed the ARI with the automated annotations and compared the ARI to the predicted pipeline rank for each of the four metrics proposed:

It can be seen there is essentially no correlation (avg. $p=0.04$), which could be for numerous reasons. For example, assigning cells to “cell types” isn’t necessarily (though admittedly often is) the end goal of clustering analysis, which can often attempt to find novel cell states. In addition, evaluating on only one (or a handful) of datasets gives no guarantee of good performance: the analysis we presented only showed these methods are good at making predictions on average. Therefore, to fully evaluate this would require running automated annotations over the full set of datasets, which in turn requires the correct selection and specification of appropriate reference atlases, as well as likely running over multiple annotation algorithms given their variable performance (<https://genomebiology.biomedcentral.com/articles/10.1186/s13059-019-1795-z>), which would significantly expand the scope of the paper.

However, given the importance of this idea and application we have added a discussion as scope for future work:

A further promising direction for future investigation is the incorporation of automated cell annotation methods as a possible metric and/or evaluation criteria. Multiple methods proposed by the community [28] take an annotated reference atlas and attempt to assign each cell in an unlabeled dataset to a previously annotated cell type based on gene expression similarity. While we have not applied such methods here due to the requirement to select an appropriate reference and annotation method, future machine learning algorithms may recommend pipelines based on agreement with automated cell type annotations or weight the deviation from automated annotation in their recommendation.

Comment 4. Since clustering resolution is regressed out of the metrics to produce the corrected metrics for downstream steps, should Figure 2A show the corrected heatmap rather than the uncorrected?

We went back and forth on this point and ultimately decided to put the uncorrected version as **Fig. 2A** with the corrected version as **Supplementary Fig. 2**, but would be happy to switch back at the reviewer's and/or editor's discretion.

Comment 5. I understand that a decision must be made to stop data collection at some point and would not suggest you need to expand the collection. But the discussion (lines 285-286) gives the impression that larger models are now limited by data availability. It may be nice to recognize recent developments like CELLxGENE, which currently has a standardized collection of 1271 datasets and >80M cells.

We thank the reviewer for raising this important point. We have updated the discussion with the following:

“However, since starting this work large-scale efforts such as CELLxGENE [24] have compiled >1200 datasets with >80M cells, raising the possibility that this benchmarking dataset may be expanded and more complex models such as deep learning may be applied.”

Comment 6. The various dataset features are largely technical and related to quality. As a discussion point, could there be opportunities incorporating additional dataset features that may better reflect characteristics of the data? Eg. features from the count matrix, like variance. Or maybe an extension of that—what if every dataset was put through some generic pipeline, several metrics get calculated (even including the ones you're optimizing for), and then the model could predict what changes to the pipeline could improve it.

This is an important point, also raised by Reviewer 1. In our reply to them (point 3), we undertook the exercise of trying to find additional dataset metadata variables that could be used for prediction. This was ultimately difficult due to the high missingness rates and the few we were able to incorporate did not meaningfully improve prediction performance. We note that our current dataset specific features are not necessarily QC

specific – we include 20 PCs of the gene expression fit across all (train) datasets, which will include biological variability (along with residual technical variability).

Comment 7. Can you confirm if FindVariableFeatures() was run following SCTransform? Typically the SCTransform() functions uses residual variance from the model to define variable genes.

Upon looking at the pipeComp source code (https://rdrr.io/bioc/pipeComp/src/inst/extdata/scrna_alternatives.R) we can confirm the FindVariableFeatures() was not run following SCTransform and the variable genes are the SCTransform returned ones.

Second round of review

Reviewer 1

I am happy the authors have largely addressed my original comments. I think clarity is still required in the methods describing the dataset-level PCA to distinguish clearly it from the single-cell PCA (having read this section of the methods over several times I see the author's intent, but a superficial read may still lead to confusion).

One new issue worth discussing briefly is the out of memory error with scran. This might be a limitation of relying on the pipeComp implementation, which doesn't appear to allow batch processing/parallelisation for scran normalisation, which would be used for larger datasets. It doesn't seem to affect the end results as far as I can tell, but it does inject missing data into the author's analysis. If anything, this highlights the further complexities of analysing large single-cell data sets, and the need for pipelines to scale appropriately with data collection.

Reviewer 2

I thank the authors for their thoughtful responses and revisions. I am satisfied and am happy to see the study published.

Reply to reviewers – Fang et al. 2024

We once again thank the reviewers and address the remaining discussion points below. Our replies are in **blue** and updates to the manuscript in **red**. We have also re-formatted the manuscript as requested.

Reviewer #1

I am happy the authors have largely addressed my original comments. I think clarity is still required in the methods describing the dataset-level PCA to distinguish clearly it from the single-cell PCA (having read this section of the methods over several times I see the author's intent, but a superficial read may still lead to confusion).

We thank the reviewer for their help in clarifying the dataset-level PCA section of the methods. To this end, we have added the following to the paragraph describing the dataset-level PCA in the methods section:

Notably, this is conceptually different from the standard PCA run on the gene by cell expression matrices in the typical scRNA-seq data analysis workflow. PCA on the dataset by gene matrix will capture the directions of variation in the mean gene expression across the 86 datasets.

One new issue worth discussing briefly is the out of memory error with scan. This might be a limitation of relying on the pipeComp implementation, which doesn't appear to allow batch processing/parallelisation for scan normalisation, which would be used for larger datasets. It doesn't seem to affect the end results as far as I can tell, but it does inject missing data into the author's analysis. If anything, this highlights the further complexities of analysing large single-cell data sets, and the need for pipelines to scale appropriately with data collection.

We appreciate the reviewer's points about injecting missing data and the need for pipelines to scale with data collection. We have added consideration of these points into the discussion section to aid researchers building on this work:

However, such datasets bring new challenges of scaling algorithms to many cells. For example, when we further attempted to assess the performance of 288 pipelines on 6 datasets with >100,000 cells we found the implementation of scan normalization used was unable to handle these data sizes, resulting in pervasive missing data. Therefore, applying such approaches to larger scRNA-seq datasets will require careful consideration and selection of highly scalable algorithms.